# Synaptotagmin-1 is the Ca²⁺ sensor for fast striatal dopamine release

**Aditi Banerjee, Jinoh Lee, Paulina Nemcova†, Changliang Liu, Pascal S Kaeser\***

Department of Neurobiology, Harvard Medical School, Boston, United States

**Abstract** Dopamine powerfully controls neural circuits through neuromodulation. In the vertebrate striatum, dopamine adjusts cellular functions to regulate behaviors across broad time scales, but how the dopamine secretory system is built to support fast and slow neuromodulation is not known. Here, we set out to identify Ca²⁺-triggering mechanisms for dopamine release. We find that synchronous dopamine secretion is abolished in acute brain slices of conditional knockout mice in which Synaptotagmin-1 is removed from dopamine neurons. This indicates that Synaptotagmin-1 is the Ca²⁺ sensor for fast dopamine release. Remarkably, dopamine release induced by strong depolarization and asynchronous release during stimulus trains are unaffected by Synaptotagmin-1 knockout. Microdialysis further reveals that these modes and action potential-independent release provide significant amounts of extracellular dopamine in vivo. We propose that the molecular machinery for dopamine secretion has evolved to support fast and slow signaling modes, with fast release requiring the Ca²⁺ sensor Synaptotagmin-1.

## Introduction

Dopamine is an important neuromodulator in the vertebrate brain, but the secretory biology of dopamine is not well understood. A key dopamine pathway arises from midbrain dopamine neurons located in the substantia nigra pars compacta. Their axons send projections to the dorsal striatum, where dopamine neuromodulation controls initiation and execution of movement. A prominent model is that dopamine operates slowly and on distant receptors through volume transmission (*Agnati et al., 1995*; *Liu and Kaeser, 2019*; *Sulzer et al., 2016*). Recent studies, however, have started to suggest that dopamine can modulate the neuronal membrane potential (*Beckstead et al., 2004*), structural synaptic plasticity (*Yagishita et al., 2014*) and behavior (*Howe and Dombeck, 2016*; *Menegas et al., 2018*) with temporal precision in the range of tens of milliseconds, suggesting the presence of molecular machines for rapid dopamine coding.

A requirement for Ca²⁺-triggering of secretion is the presence of Ca²⁺ sensors. Various Ca²⁺ binding proteins are used as Ca²⁺ sensors for vesicular exocytosis (*Kaeser and Regehr, 2014*; *Pang and Südhof, 2010*), and each could be a candidate for dopamine release. Fast synaptic transmission relies on Synaptotagmin-1, –2 or –9 (*Fernández-Chacón et al., 2001*; *Sun et al., 2007*; *Xu et al., 2007*). Synapses without these fast Synaptotagmins have prominent asynchronous release (*Turecek and Regehr, 2019*). At these asynchronous and other synapses, the higher affinity Ca²⁺ sensors Synaptotagmin-7 and Doc2, and possibly additional sensors, mediate asynchronous release (*Bacaj et al., 2013*; *Kaeser and Regehr, 2014*; *Wen et al., 2010*; *Yao et al., 2011*). At inner ear ribbon synapses, release is triggered by otoferlin, a Ca²⁺ sensor with different Ca²⁺ binding properties and kinetics (*Michalski et al., 2017*; *Roux et al., 2006*). In chromaffin cells, which release catecholamines, a major release component is left after Synaptotagmin-1 deletion, and this component is likely mediated by Synaptotagmin-7 (*Schonn et al., 2008*; *Voets et al., 2001*; *de Wit et al., 2009*). Knockdown of Synaptotagmin-1, –4 or –7 resulted in partial impairments of [³H]-dopamine released into the supernatant in response to KCl depolarization of cultured midbrain neurons, and BDNF release is also modulated by Synaptotagmin-4, but at least Synaptotagmin-4 is unlikely to operate as

**\*For correspondence:** kaeser@hms.harvard.edu

**Present address:** †Department of Neuronal Plasticity, Max Planck Institute of Psychiatry, Munich, Germany

**Competing interests:** The authors declare that no competing interests exist.

a $Ca^{2+}$ sensor in these experiments (*Dai et al., 2004*; *Dean et al., 2009*; *Mendez et al., 2011*; *Wang and Chapman, 2010*). In this study, we find that fast dopamine secretion is abolished in the striatum of mouse mutants that lack Synaptotagmin-1 in dopamine neurons, and conclude that Synaptotagmin-1 is the $Ca^{2+}$ sensor for fast dopamine release.

## Results and discussion

We here set out to identify $Ca^{2+}$-triggering mechanisms for rapid dopamine signaling. First, we analyzed the dependence of striatal dopamine release on the extracellular $Ca^{2+}$ concentration ($[Ca^{2+}]_{ex}$) in acute brain slices. We generated mice that express channelrhodopsin-2 (ChR2) selectively in dopamine neurons using mouse genetics (*Figure 1A*) and measured optogenetically evoked dopamine transients in slices of the dorsal striatum using carbon fiber amperometry (*Figure 1—figure supplement 1*). Similar to electrical stimulation paradigms (*Brimblecombe et al., 2015*; *Ford et al., 2010*), optogenetically triggered dopamine release was steeply $[Ca^{2+}]_{ex}$ dependent below 2 mM $[Ca^{2+}]_{ex}$ (*Figure 1B and C*). At 2 mM $[Ca^{2+}]_{ex}$, the 20–80% rise time was $1.91 \pm 0.13$ ms, which includes the diffusion of dopamine from the release site to the electrode, and rise times slowed down in low $[Ca^{2+}]_{ex}$ (*Figure 1D*). Together, these data establish that action potential-triggered dopamine release is mostly synchronous, and suggest the presence of a fast, low-affinity $Ca^{2+}$ sensor.

The Allen Brain Atlas and single cell sequencing data (*Lein et al., 2007*; *Saunders et al., 2018*) suggest that of the putative $Ca^{2+}$ sensors, Synaptotagmin-1 and –7 expression levels are high in midbrain dopamine neurons, while expression of the other candidates, Synaptotagmin-2 and −9, Doc2 and otoferlin, appears to be low. Using subcellular fractionation, we found that Synaptotagmin-1 is present in striatal synaptosomes that were positive for the dopamine cell marker tyrosine hydroxylase (TH) and the active zone protein Bassoon (*Figure 1—figure supplement 2*). We hypothesized that Synaptotagmin-1 is the main $Ca^{2+}$ sensor for dopamine release. We generated conditional knockout mice in which we deleted Synaptotagmin-1 from dopamine neurons (Syt-1 cKO$^{DA}$ mice, *Figure 1E*) by crossing mice with 'floxed' conditional alleles for Synaptotagmin-1 (*Skarnes et al., 2011*; *Zhou et al., 2015*) to DAT$^{IRES-Cre}$ mice (*Bäckman et al., 2006*). In these mice, we expressed oChIEF-citrine, a fast channelrhodopsin, selectively in dopamine neurons using AAVs (*Figure 1A*) to optogenetically evoke dopamine release through triggering of axonal action potentials (*Liu et al., 2018*). A 1 ms light pulse triggered dopamine release with a rise time of $1.81 \pm 0.23$ ms in control mice, but in Syt-1 cKO$^{DA}$ mice dopamine release was effectively abolished (*Figure 1F and G*). During short stimulus trains (10 stimuli at 10 Hz), dopamine release strongly depressed in Syt-1 control mice (*Figure 1H and I*), which was not due to action potential failures (*Figure 1—figure supplement 3*), but likely a consequence of the high initial release probability (*Figure 1—figure supplement 1C and D*, and *Liu et al., 2018*). In Syt-1 cKO$^{DA}$ mice, stimulus trains failed to evoke measurable dopamine release. Hence, Synaptotagmin-1 is likely the main $Ca^{2+}$ sensor that mediates synchronous dopamine release.

Striatal dopamine release is not only triggered by ascending action potentials, but also by cholinergic interneurons that innervate dopamine axons and trigger release via activation of axonal nicotinic acetylcholine receptors (nAChRs) (*Threlfell et al., 2012*; *Zhou et al., 2001*). This mechanism dominates in response to electrical stimulation in the slice preparation used here and accounts for as much as 90% of the released dopamine (*Liu et al., 2018*). We used electrical stimulation to assess whether Synaptotagmin-1 triggers dopamine release initiated by nAChR activation. Electrically evoked dopamine release was also abolished in Syt-1 cKO$^{DA}$ mice (*Figure 1—figure supplement 4*), indicating that Synaptotagmin-1 mediates both release evoked by ascending action potentials and release triggered by nAChR activation.

To assess whether loss of Synaptotagmin-1 and synchronous dopamine release has effects on striatal structure, we used 3D-structured illumination superresolution microscopy (*Gustafsson et al., 2008*; *Liu et al., 2018*). Striatal dopamine axons were labeled by TH, and their length and density were unchanged in Syt-1 cKO$^{DA}$ mice (*Figure 2A–C*). Dopamine release sites can be marked by Bassoon (*Liu et al., 2018*), and Bassoon clustering inside TH axons was not strongly affected in Syt-1 cKO$^{DA}$ mice (*Figure 2D and E*). Bassoon cluster volumes were unchanged, and there was only a very mild increase in Bassoon cluster densities in Syt-1 cKO$^{DA}$. Local shuffling of Bassoon objects decreased Bassoon density but increased the volume of the clusters artificially localized inside of TH axons in both genotypes. This confirms that Bassoon

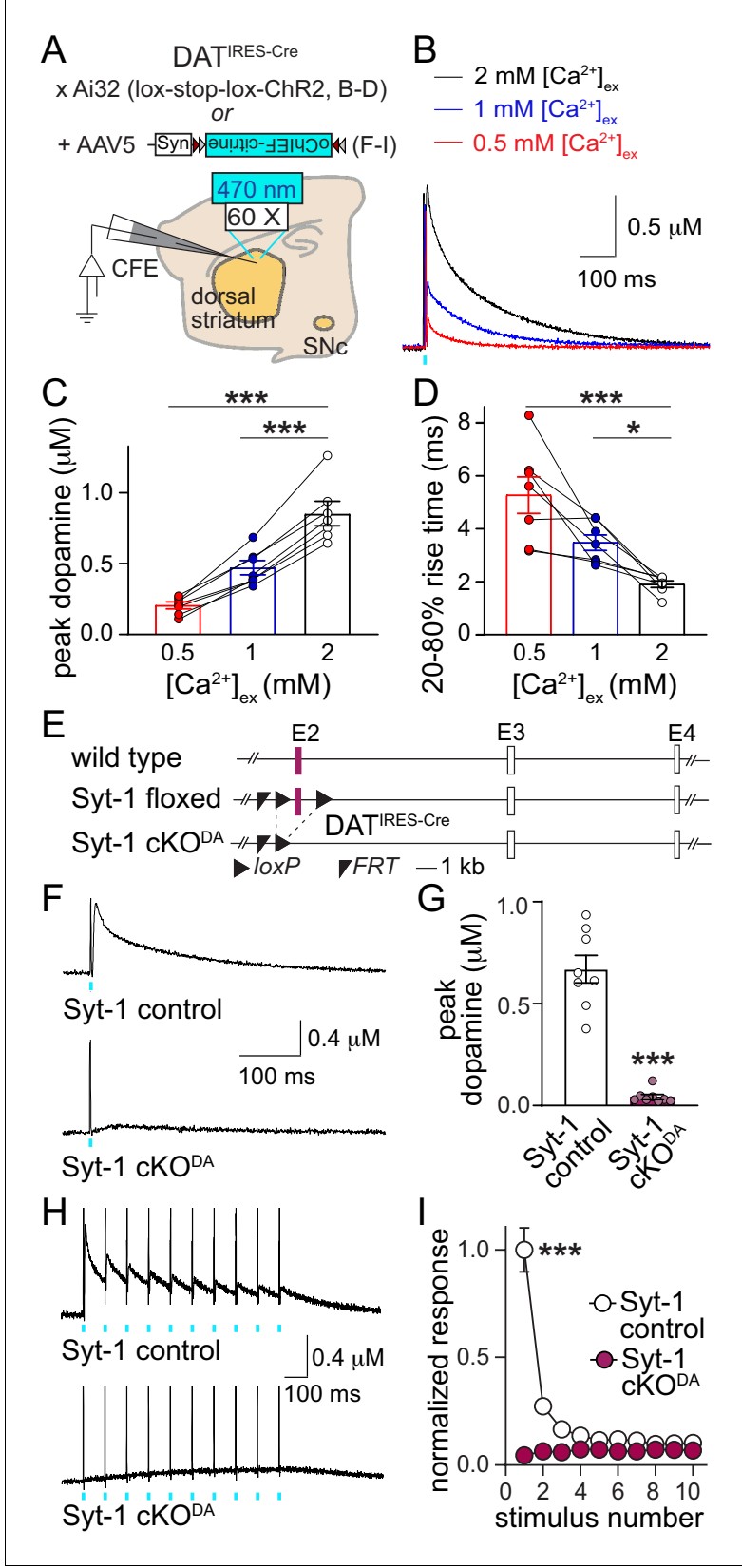

**Figure 1.** Synaptotagmin-1 is required for synchronous dopamine release. (**A**) Schematic of the experimental setup for Cre-dependent expression of channelrhodopsin variants using mutant mice (**B–D**) and AAVs (**F–I**). (**B–D**)

*Figure 1 continued*

Sample traces (B, average of four sweeps), and quantification of peak amplitudes (C) and 20–80% rise times (D) of dopamine release evoked by optogenetic activation (1 ms light pulse at 470 nm) at different $[Ca^{2+}]_{ex}$, n = 7 slices/4 mice at each $[Ca^{2+}]_{ex}$. (E) Schematic of the generation of dopamine neuron specific Synaptotagmin-1 knockout (Syt1-cKO$^{DA}$) mice. (F, G) Sample traces (F, average of four sweeps) and quantification of peak amplitudes (G) of dopamine release evoked by a 1 ms light pulse, Syt-1 control n = 8 slices/4 mice, Syt-1 cKO$^{DA}$ n = 8/4. (H, I) Sample traces (H, average of four sweeps) and quantification (I) of dopamine release evoked by ten 1 ms light pulses at 10 Hz. Amplitudes are normalized to the average first amplitude in Syt-1 control, Syt-1 control n = 8/4, Syt-1 cKO$^{DA}$ n = 8/4. All data are shown as mean ± SEM, *p<0.05, ***p<0.001. Recordings are performed in 2 mM $[Ca^{2+}]_{ex}$ unless noted otherwise, statistical significance was determined by one-way ANOVA followed by Dunnett's multiple comparisons test in C and D, by Mann-Whitney test in G, and by two-way ANOVA followed by Sidak's multiple comparisons test in I (*** for genotype, stimulus number and interaction, post-tests: *** for stimuli 1 and 2, ** for stimulus three and p>0.05 for stimuli 4–10).

The online version of this article includes the following figure supplement(s) for figure 1:

**Figure supplement 1.** Carbon fiber recordings.
**Figure supplement 2.** Synaptotagmin-1 is present in TH-positive striatal synaptosomes.
**Figure supplement 3.** Action potential firing of dopamine axons persists upon Synaptotagmin-1 knockout.
**Figure supplement 4.** Release evoked by electrical stimulation is abolished upon Synaptotagmin-1 knockout.

clusters are more frequent within TH axons than in areas surrounding these axons and that the Bassoon clusters in dopamine axons are smaller than the nearby Bassoon clusters (*Liu et al., 2018*), and indicates signal specificity. These experiments indicate that dopamine axons and release sites develop mostly normally in Syt-1 cKO$^{DA}$ mice despite the strong impairments in dopamine release.

We next tested whether strong depolarization could lead to dopamine release in the absence of Synaptotagmin-1. Local puffing of KCl onto brain slices causes a strong depolarization of dopamine axons and surrounding neurons, for example cholinergic interneurons, which triggers massive dopamine release that requires the active zone protein RIM in dopamine axons (*Liu et al., 2018*). Compellingly, puffing KCl directly onto the recording site evoked an amperometric response that was indistinguishable between control and Syt-1 cKO$^{DA}$ mice (*Figure 2F–I*). This establishes that dopamine is produced, loaded into vesicles, and released by strong depolarization throughout a structurally largely normal dopamine axon in Syt-1 cKO$^{DA}$ mice. While the exact mechanism of KCl depolarization induced dopamine release is not known, the data further suggest that depolarization-induced, likely massive $Ca^{2+}$ entry may trigger vesicular dopamine release via one or multiple alternative $Ca^{2+}$ sensors in the absence of Synaptotagmin-1.

These additional sensors may for example mediate asynchronous release, a form of vesicular exocytosis at synapses that is triggered with a longer, variable delay in response to action potentials and $Ca^{2+}$ entry (*Kaeser and Regehr, 2014*; *Pang and Südhof, 2010*). However, if asynchronous dopamine release is present in Syt-1 cKO$^{DA}$ mice, the amount of dopamine after one or 10 action potentials is too small to be detected (*Figure 1F–I*). It is possible that the dopamine transporter (DAT) is sufficient to mediate re-uptake of dopamine before extracellular accumulation is observed. To test this hypothesis, we repeated the optogenetic experiments with 10 Hz stimulus trains before and after wash-in of the DAT-blocker nomifensine. In control slices, nomifensine resulted in a reduction of the first amplitude in the train (*Figure 3A and B*), which may be due to suppression mediated by tonic axonal D2-receptor activation (*Benoit-Marand et al., 2001*; *Ford, 2014*). When we assessed release throughout the train (measured as the area under the curve), a robust enhancement was observed in Syt-1 control and Syt-1 cKO$^{DA}$ mice (*Figure 3C*), and the total increase in extracellular dopamine upon DAT blockade was similar between Syt-1 control and Syt-1 cKO$^{DA}$ (*Figure 3D*). When normalized to the measured extracellular dopamine before DAT blockade, it amounted to 1.7-fold and 31.5-fold enhancements in Syt-1 control and cKO$^{DA}$ mice, respectively (*Figure 3E*). These experiments indicate that asynchronous release is present and persists in Syt-1 cKO$^{DA}$ mice, and this can be detected upon blockade of dopamine re-uptake. While sensors for asynchronous

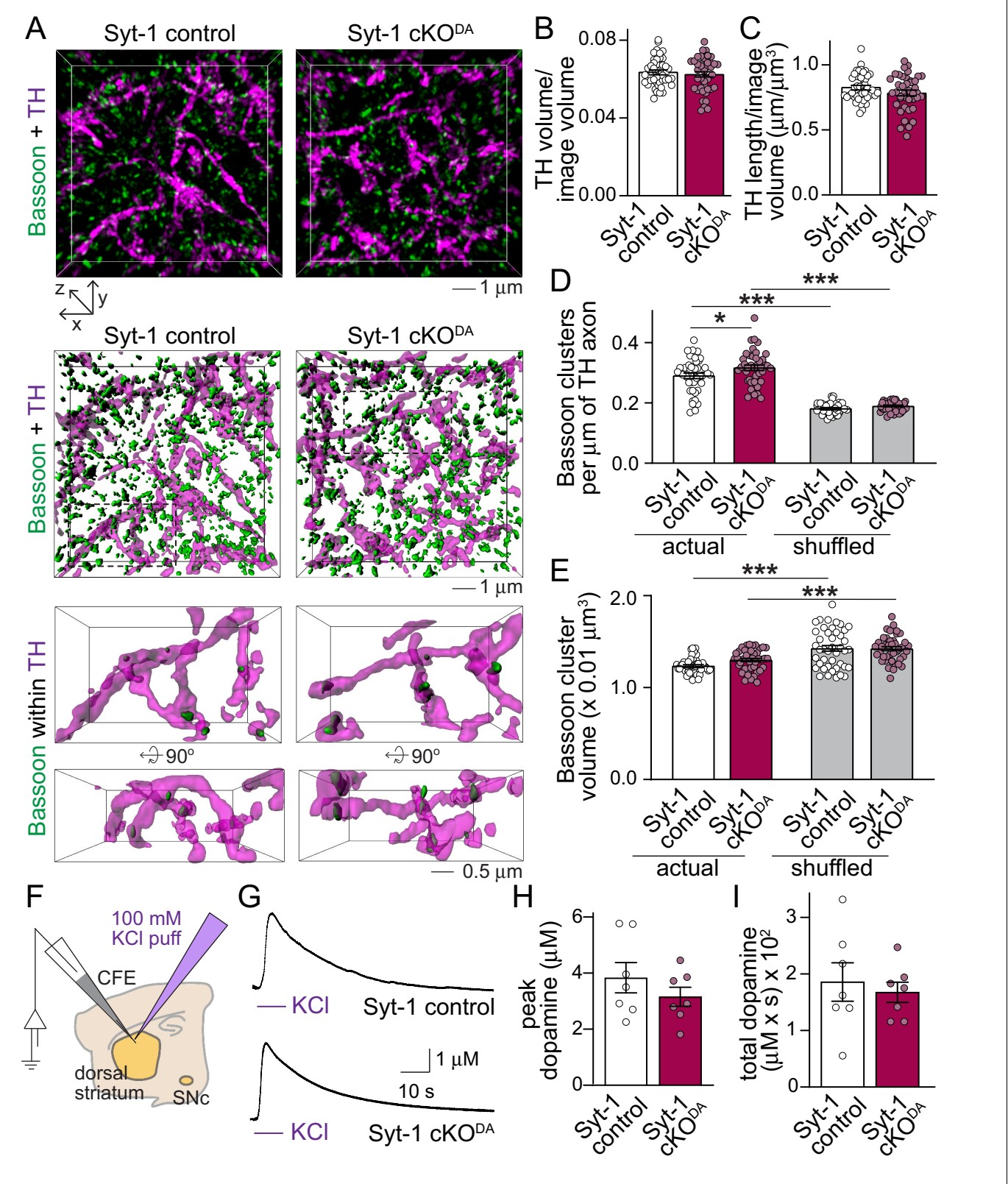

**Figure 2.** Dopamine axon structure and depolarization induced dopamine release are intact after ablation of Synaptotagmin-1. (**A**) Sample 3D-SIM images of striatal brain sections of Syt-1 control and Syt-1 cKO^DA mice stained for the dopamine axon marker TH and the release site marker Bassoon. Volume rendered reconstructions (10 × 10 × 2 μm³, top, all Bassoon is shown), surface rendering of the same volumes (middle, all Bassoon is shown),
*Figure 2 continued on next page*

*Figure 2 continued*

and zoomed-in volumes (bottom, $5 \times 3 \times 2 \ \mu m^3$ in front view and rotated by $+90°$ along the x-axis, only including Bassoon clusters with >40% volume overlap with TH) are shown. (B–E) Quantification of the fraction of the image volume covered by TH (B), TH axon length (C), Bassoon cluster densities (D) and Bassoon cluster volumes (E). For the shuffled controls in D and E, each Bassoon object was randomly relocated 1000 times within a volume of $1 \times 1 \times 1 \ \mu m^3$, and the actual Bassoon densities and volumes were compared to the average of the shuffled controls. Syt-1 control: n = 43 images/5 slices/3 mice, Syt-1 cKO$^{DA}$ n = 41/5/3. (F–I) Schematic of the experiment (F), sample traces (G) and analyses of peak amplitudes (H) and total dopamine (I, quantified as area under the curve) of dopamine release measured in response to puffing of 100 mM KCl onto the recording area. Syt-1 control: n = 7 slices/3 mice, Syt-1 cKO$^{DA}$ n = 7/3. All data are shown as mean ± SEM, *p<0.05, ***p<0.001, statistical significance was determined by unpaired t tests in B and C, one-way ANOVA followed by Sidak's multiple comparisons test in D and E, and Mann-Whitney tests in H and I .

dopamine release are not known, the presence of Synaptotagmin-7 in substantia nigra dopamine neurons and its role in asynchronous release at fast synapses (*Bacaj et al., 2015*; *Lein et al., 2007*; *Mendez et al., 2011*; *Saunders et al., 2018*; *Wen et al., 2010*) makes Synaptotagmin-7 a candidate sensor protein.

In vivo, asynchronous release may significantly contribute to extracellular dopamine. To test this hypothesis, we performed microdialysis in anesthetized mice. In Syt-1 control mice, extracellular dopamine levels were reduced to approximately one third upon reverse dialysis of the sodium channel blocker tetrodotoxin (TTX), which inhibits action potential firing (*Figure 3F and G*). Remarkably, extracellular dopamine levels before TTX reverse dialysis were only mildly reduced in Syt-1 cKO$^{DA}$ mice and action potential blockade robustly reduced extracellular dopamine in these mice. Hence, asynchronous release may provide a substantial amount of extracellular dopamine in vivo. These data suggest that three modes of dopamine release exist: synchronous and asynchronous release in response to action potentials, and action potential independent release that is likely mediated by spontaneous exocytotic events. Remarkably, each component appears to contribute significantly to the extracellular dopamine measured by microdialysis in anesthetized mice, suggesting that asynchronous and action potential-independent release may be prominent. It is not known whether Synaptotagmin-1 knockout affects spontaneous dopamine release, but literature from conventional synapses establishes that Synaptotagmin-1 knockout strongly enhances miniature synaptic vesicle release (*Broadie et al., 1994*; *Xu et al., 2009*). Hence, it is possible that Syt-1 cKO$^{DA}$ leads to increased spontaneous dopamine release in the striatum. But the observation that extracellular dopamine levels after TTX are not increased in Syt-1 cKO$^{DA}$ mice suggests that there is no dramatic enhancement of miniature dopamine release, that microdialysis is not sufficiently sensitive to detect such a change, or that an enhancement is counteracted by dopamine clearance.

Here, we find that synchronous striatal dopamine release requires the fast $Ca^{2+}$ sensor Synaptotagmin-1. Given the prevailing model of volume transmission, which postulates that dopamine neurotransmission is slow and imprecise, it is surprising that dopamine neurons have evolved to employ a fast $Ca^{2+}$ sensor. This finding, however, is in line with a recent description of sparse secretory hotspots in striatal dopamine axons (*Liu et al., 2018*). We propose that at these sparse sites, dopamine is rapidly and synchronously released with a high vesicular release probability to generate an extracellular dopamine signal that is spatially restricted and has rapid kinetics. While the dopamine receptor distribution relative to the dopamine release hotspots is not known, our data generally suggest that dopamine transmission may be fast and compartmentalized within a target cell. These mechanisms may support fast dopamine coding functions (*Howe and Dombeck, 2016*; *Menegas et al., 2018*; *Yagishita et al., 2014*). Our work further suggests that slower signaling modes exist. Substantial amounts of extracellular dopamine may come from asynchronous and action potential-independent dopamine release. Future studies should address which dopamine functions rely on fast dopamine signaling machinery, and whether some functions are supported by slower signaling mechanisms.

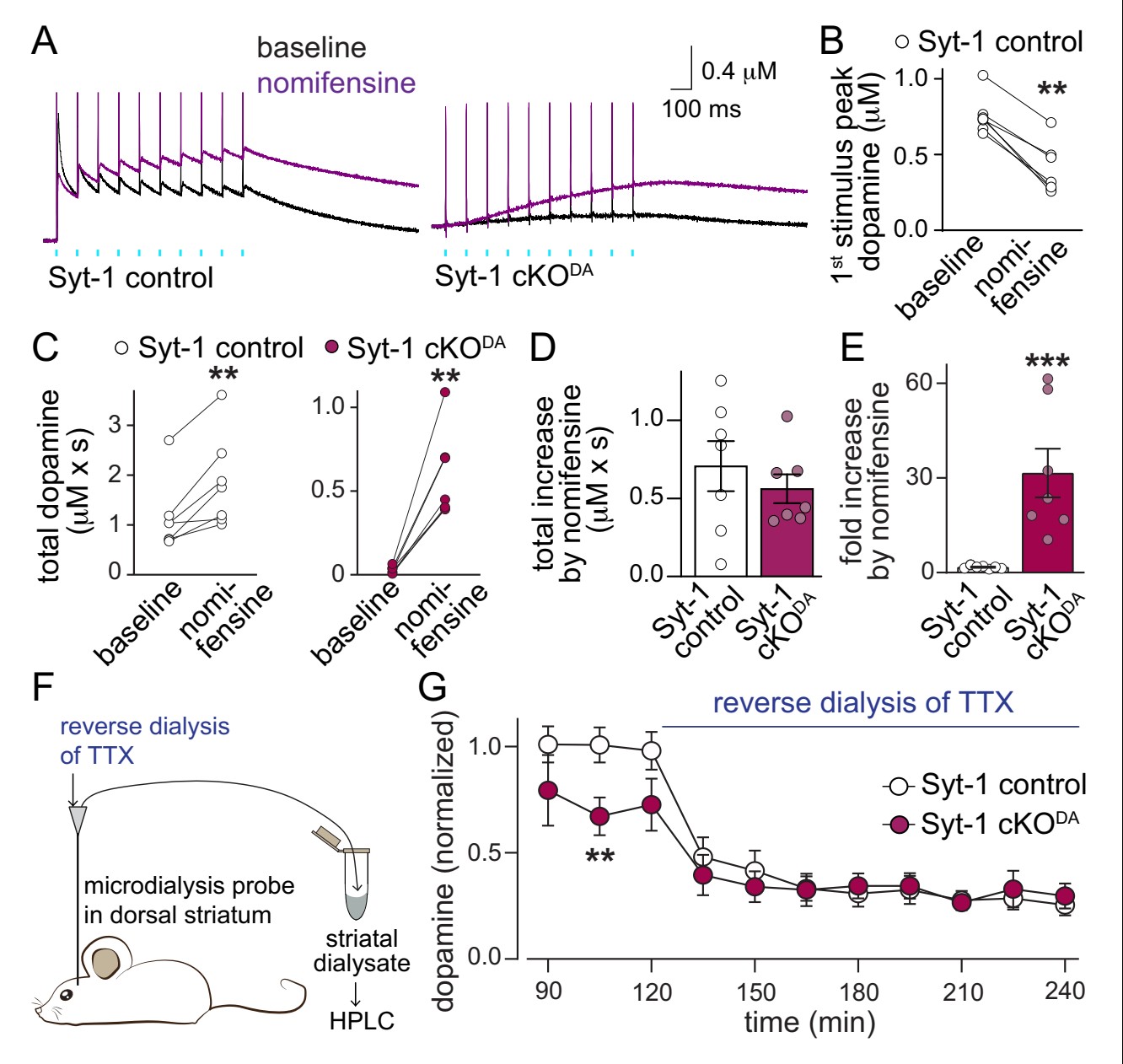

**Figure 3.** Asynchronous dopamine release sustains extracellular dopamine in vivo after ablation of Synaptotagmin-1. (A–E) Sample traces (A, average of four sweeps) and quantification of dopamine release (B–E) evoked by a 10 Hz stimulus train induced as described in *Figure 1H* before (black traces) and after addition of the DAT blocker nomifensine (10 µM, purple traces) in Syt-1 control and Syt-1 cKO$^{DA}$ slices. Amplitudes of the first response (B, Syt-1 control only), total dopamine (area under the curve for 2.935 s after the 1$^{st}$ stimulus) before and after nomifensine (C), subtracted area (D) and fold increase after nomifensine (E) are shown, Syt-1 control n = 7 slices/5 mice, Syt-1 cKO$^{DA}$ n = 7/5. (F, G) Schematic of the experiment (F) and summary plot (G) of in vivo dopamine measurements using microdialysis in the dorsal striatum (microdialysates were collected over periods of 15 min and values measured in these microdialysates at the end of each period are plotted) of Syt-1 control and Syt-1 cKO$^{DA}$ mice. The quantification in G shows dopamine levels normalized to the average concentration from the 76$^{th}$ - 120$^{th}$ min in Syt-1 control, and reverse dialysis of 10 µM TTX to block action potential firing started at 121 min, Syt-1 control n = 5 mice, Syt-1 cKO$^{DA}$ n = 5 mice. All data are shown as mean ± SEM, **p<0.01, ***p<0.001, statistical significance was determined by Wilcoxon matched pairs signed rank tests in B and C, Mann Whitney tests in D and E, and by two-way ANOVA followed by Sidak's multiple comparisons test in G. In G, the data followed a lognormal distribution and statistical testing was done after the data were converted to a log$_e$ scale, *** for time and ** for interaction, post-tests for genotypes are shown.

# Materials and methods

## Key resources table

| Reagent type (species) or resource | Designation | Source or reference | Identifiers | Additional information |
|---|---|---|---|---|
| Genetic reagent (*M. musculus*) | B6.SJL-Slc6a3[tm1.1(cre)Bkmm]/J; DAT[IRES-Cre] | *Bäckman et al., 2006* | JAX 006660, RRID:IMSR_JAX:006660 | |
| Genetic reagent (*M. musculus*) | B6.129S-Gt(ROSA)26Sor[tm32(CAG-COP4*H134R/EYFP)Hze]; Ai32 | *Madisen et al., 2012* | JAX 012569, RRID:IMSR_JAX:012569 | |
| Genetic reagent (*M. musculus*) | C57BL/6NTac-[Syt1tm1a(EUCOMM)Wtsi]/WtsiCnrm; Syt-1 floxed; *Syt1[lox/lox]* | *Skarnes et al., 2011*; *Zhou et al., 2015*; obtained from Dr. T.C. Südhof | EUCOMM (EM:06829), RRID:IMSR_EM:06829 | Conditional Synaptotagmin-1 floxed mice |
| Cell line (*H. sapiens*) | HEK293T | ATCC | Cat#: CRL-3216, RRID:CVCL_0063 | |
| Recombinant DNA reagent | AAV-flex-oChIEF-citrine | Addgene; *Lin et al., 2009* | Plasmid# 50973, RRID:Addgene_50973 | |
| Chemical compound, drug | Nomifensine | Tocris | Cat. No. 1992 | |
| Chemical compound, drug | Tetrodotoxin | Tocris | Cat. No. 1078 | |
| Antibody | Mouse monoclonal IgG2a anti-Bassoon SAP7F407 (A85) | Enzo Life Sciences | Cat# ADI-VAM-PS003-F, RRID:AB_11181058 | IHC (1:500) |
| Antibody | Guinea pig polyclonal anti-Tyrosine hydroxylase (A111) | Synaptic Systems | Cat# 213 104, RRID:AB_2619897 | IHC (1:1000) |
| Antibody | Rabbit polyclonal anti-Synaptotagmin-1 antiserum (A24) | gift from Dr. T.C. Südhof | V216 | ICC (1:1000) |
| Other | Microdialysis probe | Harvard Apparatus | Item# CMA8309581 | |
| Other | Carbon fiber filaments | Goodfellow | Item# C 005722 | |
| Software, algorithm | Fiji | *Schindelin et al., 2012* | RRID:SCR_002285, https://imagej.net/Fiji/Downloads | Used for confocal synaptosome and 3D-SIM slice experiments |
| Software, algorithm | SoftWoRX | GE Healthcare | http://incelldownload.gehealthcare.com/bin/download_data/SoftWoRx/7.0.0/SoftWoRx.htm | Used for 3D reconstruction |
| Software, algorithm | Custom MATLAB code | *Liu et al., 2018* | https://github.com/hmslcl/3D_SIM_analysis_HMS_Kaeser-lab_CL (*Liu, 2017*) | Used for analysis of 3D-SIM and synaptosome images |
| Software, algorithm | Prism8 | GraphPad | RRID:SCR_002798, https://www.graphpad.com/scientific-software/prism | Used for statistical analysis |

## Mice

DAT[IRES-Cre] mice (*Bäckman et al., 2006*) express Cre recombinase under the dopamine transporter (DAT) gene locus, and were obtained from the Jackson laboratories (RRID:IMSR_JAX: 006660, B6.SJL-Slc6a3[tm1.1(cre)Bkmm]/J). Mice for the Cre-dependent expression of ChR2 (*Madisen et al., 2012*) (Ai32, RRID:IMSR_JAX:012569, B6.129S-Gt(ROSA)26Sor[tm32(CAG-COP4*H134R/EYFP)Hze]), obtained from

the Jackson Laboratories, were crossed to DAT[IRES-Cre] mice, and mice used for experiments were heterozygote for both DAT[IRES-Cre] and ChR2. The conditional Synaptotagmin-1 knockout mice were generated by EUCOMM (EM:RRID:IMSR_EM:06829, C57BL/6NTac-[Syt1tm1a(EUCOMM)Wtsi]/WtsiCnrm) (*Skarnes et al., 2011*), described in *Zhou et al., 2015*, and obtained from Dr. T.C. Südhof. Syt-1 cKO[DA] mice were generated by crossing Syt-1 floxed mice with DAT[IRES-Cre] mice, and for all experiments, Syt-1 cKO[DA] mice were mice homozygote for the Syt-1 floxed allele and heterozygote for DAT[IRES-Cre]. Syt-1 control mice were siblings of Syt-1 cKO[DA] mice with two wild type Syt-1 alleles and a heterozygote DAT[IRES-Cre] allele, except for KCl puffing experiments in *Figure 2F–I* and electrical stimulation experiments in *Figure 1—figure supplement 4*. In these experiments, Syt-1 controls were either heterozygote for the Syt-1 floxed allele and for DAT[IRES-Cre], or homozygote for the Syt-1 floxed allele without a DAT[IRES-Cre] allele. Mice were group housed in a 12 hr light-dark cycle with free access to water and food, and experiments were done in male and female mice. All animal experiments were done in accordance with approved protocols of the Harvard University Animal Care and Use Committee.

## Production of AAV viruses and stereotaxic surgeries

In Syt-1 control and Syt-1 cKO[DA] mice striatal dopamine fibers were activated after transduction of dopamine neurons with AAVs for Cre-dependent expression of oChIEF-citrine (*Lin et al., 2009*), a fast channelrhodopsin variant (p867, RRID:Addgene_50973). AAVs (serotype AAV2/5) were generated in HEK293T cells (purchased as identified, mycoplasma free cell line from ATCC, Cat#: CRL-3216, RRID:CVCL_0063) using calcium phosphate transfection. 72 hr after transfection, cells were collected, lysed, and viral particles were extracted and purified from the 40% layer after iodixanol gradient ultracentrifugation. Quantitative rtPCR was used to measure the genomic titer ($2.31$–$2.75 \times 10^{12}$ genome copies/ml). For stereotaxic surgeries, mice were anesthetized using 5% isoflurane and mounted on a stereotaxic frame. 1.5–2% isoflurane was used to maintain a stable anesthesia during the surgery. 1 µl of viral solution was injected unilaterally into the right substantia nigra pars compacta (SNc – 0.6 mm anterior, 1.3 mm lateral of Lambda and 4.2 mm below pia) of Syt-1 control and Syt-1 cKO[DA] mice at P25-29 using a microinjector (PHD ULTRA syringe pump, Harvard Apparatus) at the rate of 100 nl/min. After surgery, the mice obtained analgesia and were allowed to recover for at least 21 d prior to recording. Stereotaxic surgeries were performed according to protocols approved by the Harvard University Animal Care and Use Committee.

## Electrophysiological recordings

Male and female mice (42–113 days old) were deeply anesthetized with isoflurane and decapitated. 250 µm thick sagittal brain sections containing the striatum were cut using a vibratome (Leica, VT1200s) in ice-cold cutting solution with (in mM): 75 NaCl, 75 sucrose, 2.5 KCl, 7.5 $MgSO_4$, 26.2 $NaHCO_3$, 1 $NaH_2PO_4$, 12 glucose, 1 sodium ascorbate, 1 myo-inositol, 3 sodium pyruvate, (pH 7.4, 300–310 mOsm). Slices were incubated at room temperature for 1 hr in incubation solution bubbled with 95% $O_2$ and 5% $CO_2$ containing (in mM): 126 NaCl, 2.5 KCl, 1.3 $MgSO_4$, 2 $CaCl_2$, 26.2 $NaHCO_3$, 1 $NaH_2PO_4$, 12 glucose, 1 sodium ascorbate, 1 myo-inositol, 3 sodium pyruvate (pH 7.4, 305–310 mOsm). Recording was done at 34–36°C, and slices were continuously perfused with artificial cerebrospinal fluid (ACSF) at 3–4 ml/min bubbled with 95% $O_2$ and 5% $CO_2$. ACSF contained (in mM): 126 NaCl, 2.5 KCl, 2 $CaCl_2$ (unless noted otherwise), 1.3 $MgSO_4$, 1 $NaH_2PO_4$, 12 glucose, 26.2 $NaHCO_3$, pH 7.4, 300–310 mOsm. Recordings were completed within 5 hr of slicing. In *Figure 3A–E*, 10 µM nomifensine (Tocris, Catalogue No.#1992) was applied to block the dopamine transporter (DAT). For all genotype comparisons, each littermate pair was recorded on the same day with interleafed recordings, and the experimenter was blind to genotype throughout recording and data analyses. All data acquisition and analyses for electrophysiology was done using pClamp10 (Clampex, Axon Instruments).

For all carbon fiber amperometry, carbon fiber microelectrodes (CFEs, 7 µm diameter, 100–150 µm long) were made from carbon fiber filaments (Goodfellow). Each CFE was calibrated by puffing freshly made dopamine solutions of increasing concentrations (0, 1, 5, 10, 20 µM) in ACSF for 10 s (*Figure 1—figure supplement 1A*). The currents for each concentration of dopamine were plotted against the dopamine concentration and only CFEs with a linear relationship were used. On each day, a new CFE was calibrated and dopamine release was measured with the same CFE for a Syt-1

control and Syt-1 cKO$^{DA}$ littermate pair. CFEs were held at 600 mV and placed 20–60 µm below the slice surface in the dorsolateral striatum. Signals were sampled at 10 kHz and low-pass filtered at 400 Hz. Dopamine release was evoked by electrical or optogenetic stimulation every 2 min.

Optogenetic stimulation was performed with channelrhodopsin expression limited to dopamine neurons by AAV-mediated expression of Cre-dependent oChIEF-citrine (*Figure 1F–I*, *Figure 3A–E*, *Figure 1—figure supplement 3B–3G*) or transgenic mice for Cre-dependent ChR2 expression (*Madisen et al., 2012*; *Figure 1B–D*, *Figure 1—figure supplement 1C and D*). For stimulation, brief 1 ms pulses of 470 nm light were delivered at the recording site in the dorsolateral striatum through a 60 x objective by a light-emitting diode (Cool LED pE4000). Optogenetic stimulation was applied as a single stimulus (*Figure 1B–D*, *Figure 1—figure supplement 1C and D* using ChR2, *Figure 1F and G* using oChIEF) or as stimulus trains (10 stimuli at 10 Hz in *Figures 1H, I* and *3A–E*, *Figure 1—figure supplement 3B–3E* or 40 stimuli at 40 Hz in *Figure 1—figure supplement 3F and G* using oChIEF). Optogenetic stimulation was applied every 2 min for all dopamine release measurements, or every 10 s for the field recordings shown in *Figure 1—figure supplement 3*. Total dopamine release during stimulation trains (*Figure 3A–E*) was measured as area under the curve from the start of the 1$^{st}$ stimulus for 2.935 s and expressed as µM x s after removal of the stimulus artefacts. The total increase (*Figure 3D*) was calculated by subtracting the area (area$_{nomifensine}$ − area$_{baseline}$), and the fold increase (*Figure 3E*) was calculated by division (area$_{nomifensine}$/area$_{baseline}$).

Electrical stimulation was performed with an ACSF filled glass pipette (tip diameter 3–5 µm) connected to a linear stimulus isolator (A395, World Precision Instruments) to deliver monopolar electrical stimulation (10–90 µA). The stimulation pipette was placed 20–30 µm below the slice surface in the dorsolateral striatum and 100–120 µm away from the tip of the CFE. A biphasic wave (0.25 µs in each phase) was applied to evoke dopamine release. Electrical stimulation was delivered either as a single stimulus or a 10-stimulus 10 Hz train (*Figure 1—figure supplement 4*).

For KCl stimulation in *Figure 2F–I*, KCl solution containing (in mM) 100 KCl, 50 NaCl, 1.3 MgSO$_4$, 2 CaCl$_2$, 12 glucose, 10 HEPES, pH 7.3, 300–310 mOsm) was puffed onto the recording site in the dorsolateral striatum for 10 s at 9 µl/s using a syringe pump (World Precision Instruments). Only one KCl puff was applied per slice. The peak amplitude of the dopamine response and the area under the curve were quantified from start of application of KCl to 200 s after the puff, at which time the levels returned to baseline.

For experiments in variable [Ca$^{2+}$]$_{ex}$, CFEs were first tested with a 20 µM dopamine puff at different [Ca$^{2+}$]$_{ex}$ (0.5, 1, 2 and 4 mM) to ensure that the CFE correctly reports [dopamine] across variable amounts of [Ca$^{2+}$]$_{ex}$ (*Figure 1—figure supplement 1B*). Slices were either recorded with decreasing [Ca$^{2+}$]$_{ex}$ (2, 1, and 0.5 mM, *Figure 1B–D*) or with increasing [Ca$^{2+}$]$_{ex}$ (2, 4 mM, *Figure 1—figure supplement 1C and D*) in separate experiments. Extracellular magnesium was adjusted for recordings in variable [Ca$^{2+}$]$_{ex}$ (solutions contained Ca$^{2+}$/Mg$^{2+}$ in mM: 0.5/2.8; 1/2.3; 2/1.3 and 4/0).

Extracellular recordings in *Figure 1—figure supplement 3* were performed with an ACSF filled glass pipette (2–3 µm tip diameter) that was placed 20–60 µm below the slice surface in areas of the dorsolateral striatum with uniform citrine fluorescence. Optogenetic stimulation was applied as a 10 Hz train (*Figure 1—figure supplement 3B–3E*) or a 40 Hz train (*Figure 1—figure supplement 3F and G*) every 10 s and 100 sweeps were averaged for quantification (*Figure 1—figure supplement 3B–3G*). Sodium channels were blocked using 1 µM TTX (Tocris, Catalogue No.# 1078) and extracellular potentials evoked by 10 Hz trains were recorded before and after TTX. To quantify the reduction by TTX, the amplitude evoked by the 1$^{st}$ stimulus in the 10 Hz train before and after TTX was analyzed in Syt-1 control and Syt-1 cKO$^{DA}$ mice (*Figure 1—figure supplement 3C and D*).

## Immunostaining of brain sections

Male and female Syt-1 control and cKO$^{DA}$ littermate mice (99–111 days old) were deeply anesthetized with 5% isoflurane and perfused transcardially with ice-cold 30–50 ml phosphate buffer saline (PBS), followed by 50 ml of 4% paraformaldehyde (PFA) in PBS at 4°C. Brains were then left in 4% PFA for 12–16 hr followed by incubation in 30% sucrose + 0.1% sodium azide in PBS overnight or until they sank to the bottom of the tube. Coronal striatal sections (20 µm thick) were cut using a vibratome (Leica, VT1000s) in ice-cold PBS. Antigen retrieval was performed on slices overnight at 60$^0$C in 150 mM NaCl, 1 mM EDTA, 0.05% Tween 20, 10 mM Tris Base, pH 9.0. After antigen retrieval, slices were washed in PBS, and incubated in Image-iT FX signal enhancer (Invitrogen, I36933) for 30 min at room temperature. Slices were washed in PBS for 10 min and non-specific

binding was blocked in 10% goat serum in 0.25% Triton X-100 in PBS (PBST) for 1 hr at room temperature. Slices were stained with primary antibodies for 12 hr at 4°C, and the following primary antibodies were used: mouse monoclonal IgG2a anti-Bassoon (1:500, A85, RRID:AB_11181058) and guinea pig polyclonal anti-TH (1:1000, A111, RRID:AB_2619897). Next, slices were washed three times in PBST for 10 min and incubated in secondary antibodies (1:500, goat anti-mouse IgG2a Alexa 488, S8, RRID:AB_2535771, and goat-anti guinea pig Alexa 568, S27, RRID:AB_2534119) for 2 hr at room temperature in PBST. Sections were washed three times in PBST for 10 min and then mounted on Poly-D-lysine coated #1.5 cover glasses (GG-18–1.5-pdl, neuVitro) with H-1000 mounting medium (Vectashield). At all times, the experimenter was blind to the genotype of the mice.

## 3D-SIM image acquisition and analysis

Image acquisition and analyses were done essentially as described before (*Liu et al., 2018*) using a DeltaVision OMX V4 Blaze structured illumination microscope (GE Healthcare) with a 60 x, 1.42 N.A. oil immersion objective and Edge 5.5 sCMOS cameras (PCO) for each channel. Z stacks were acquired with 125 nm step size and 15 raw images per plane (five phases, three angles). Immersion oil matching was used to minimize spherical aberration. Lateral shift between green and red channels was measured using a control slide to generate a calibration image and all images were reconstructed using this calibration to remove lateral shifts. All raw images were aligned and reconstructed to obtain superresolved images using the image registration function in softWoRx. Image volumes ($40 \times 40 \times 6$ µm$^3$) were acquired from 7 to 8 regions within the dorsolateral striatum in each section. For image analysis, regions of interest (ROIs) ranging from $20 \times 20 \times 2.5$ µm$^3$ to $25 \times 25 \times 2.5$ µm$^3$ were selected manually in each image stack. ROIs were analyzed to characterize TH and Bassoon signals and to determine their overlap using a custom written MATLAB code (*Liu et al., 2018*) (available at https://github.com/hmslcl/3D_SIM_analysis_HMS_Kaeser-lab_CL; *Liu, 2017*). Briefly, intensity thresholding using Otsu and size thresholding (0.04–20 µm$^3$ for TH axons, 0.003–0.04 µm$^3$ for Bassoon) were applied to each ROI. The volume occupied by TH was quantified and was divided by the total image volume (*Figure 2B*). The TH signals were skeletonized to determine TH axon length by 3D Gaussian filtering and a homotypic thinning algorithm. TH axon length was divided by the total image volume in Syt-1 control and Syt-1 cKO$^{DA}$ mice (*Figure 2C*). The volume of Bassoon clusters and percentage overlap of Bassoon with TH was calculated, and >40% overlap of Bassoon with TH was considered to be a positive association (*Liu et al., 2018*; *Figure 2D and E*). For generating controls using local shuffling, each Bassoon object was randomly shuffled within $1 \times 1 \times 1$ µm$^3$ for 1000 rounds of shuffling and for each ROI, and the overlap between the average of shuffled Bassoon and TH was calculated. Sample images in *Figure 2A* were generated using Imaris 9.0.2 (Oxford Instruments) from masked images of either 'Bassoon + TH' or 'Bassoon within TH' derived from the custom written MATLAB code. Adjustments of contrast, intensity and surface rendering were done identically for each condition for illustration, but after quantification. For all 3D-SIM data acquisition and analyses, the experimenter was blind to the genotype of the mice.

## Striatal synaptosome preparation and immunostaining

Striatal synaptosome preparations were performed as previously described (*Liu et al., 2018*). Wild type mice (P21-70) were deeply anesthetized, decapitated, and brains were harvested into ice-cold PBS. Dorsal striata were dissected and placed into a pre-cooled detergent-free glass tube and 1 ml of ice-cold homogenizing buffer containing (in mM): 4 4-(2-hydroxyethyl)−1-piperazineethanesulfonic acid (HEPES), 320 sucrose, pH 7.4, and 1x of a mammalian protease inhibitor cocktail was added. A detergent-free ice-cold glass-teflon homogenizer was used to homogenize the tissue using 12 strokes. The striatal homogenate was added to 1 ml of homogenizing buffer and centrifuged at 1,000 g for 10 min at 4°C. The supernatant (S1) was collected and centrifuged at 12,500 g for 15 min at 4°C. The supernatant (S2) was removed and the pellet (P2) was re-homogenized in 1 ml homogenizing buffer with six strokes. A sucrose density gradient was prepared with 5 ml of both 0.8 M and 1.2 M sucrose in thin wall ultracentrifugation tubes (Beckman Coulter, Cat # 344059). The P2 homogenate was mixed with 1 ml of homogenizing buffer, and 1.5 ml was added to the top of the sucrose gradient and was centrifuged at 69,150 x g for 70 min at 4°C (SW 41 Ti Swinging-Bucket Rotor, Beckman Coulter, Cat. # 331362). The synaptosome layer (1–1.5 ml) was collected from the interface of

the two sucrose layers. Synaptosomes were then diluted 20–30 times in homogenizing buffer and spun (4000 x g, 10 min) onto Poly-D-lysine coated #1.5 coverslips at 4°C. Excess homogenizing buffer was pipetted out and synaptosomes were fixed using 4% PFA in PBS for 20 min at 4°C. Non-specific binding block and permeabilization was done in 3% bovine serum albumin + 0.1% Triton X-100 in PBS at room temperature for 45 mins. Primary antibody staining was done for 12 hr at 4°C, followed by three washes for 15 mins each. Secondary antibody staining was done for 2 hr at room temperature in blocking solution followed by three washes each for 15 mins. The primary antibodies used were: mouse monoclonal IgG2a anti-Bassoon (1:1000, A85, RRID:AB_11181058), guinea pig polyclonal anti-TH (1:1000, A111, RRID:AB_2619897), and rabbit polyclonal anti-Synaptotagmin-1 antiserum (1:1000, A24, V216, a gift from Dr. T.C. Südhof). The secondary antibodies were: goat anti-mouse IgG2a Alexa 488 (1:500, S8, RRID:AB_2535771), goat anti-rabbit Alexa 555 (1:500, S22, RRID:AB_2535849), and goat anti-guinea pig Alexa 633 (1:500, S34, RRID:AB_2535757).

## Confocal microscopy and image analysis of striatal synaptosomes

Single optical sections of synaptosomes ($105 \times 105$ $\mu m^2$) stained for Bassoon (detected via Alexa 488), Synaptotagmin-1 (detected via Alexa-555) and TH (detected via Alexa 633) were imaged with an oil immersion 60 x objective and 1.5 x optical zoom using an Olympus FV1000 confocal microscope. For quantification, images were processed for background subtraction in Fiji using the 'rolling ball' algorithm with a radius of 1 $\mu m$ for each channel. Each background subtracted image was analyzed in a custom MATLAB program (available at https://github.com/hmslcl/3D_SIM_analysis_HMS_Kaeser-lab_CL; *Liu, 2017*). Otsu intensity thresholds, size thresholds (0.2–1 $\mu m^2$) and shape thresholds (ratio of x to y axis <1.5) were applied for object detection. These threshold settings were identical for each image and allowed for unbiased and automated detection of Bassoon-positive (Bassoon$^+$), TH-positive (TH$^+$) and Synaptotagmin-1-positive (Syt-1$^+$) ROIs in each image, with a total of 300–600 synaptosome objects detected per image. These ROIs were then used to generate the Bassoon$^+$TH$^+$, Bassoon$^-$TH$^+$, Bassoon$^+$TH$^-$ ROIs displayed in *Figure 1—figure supplement 2C*. The extent of overlap of Synaptotagmin-1$^+$ ROIs with the various Bassoon/TH ROIs was quantified and a 20–100% overlap criterion was applied to define Synaptotagmin-1 positivity for the various Bassoon/TH ROIs. Sample images in *Figure 1—figure supplement 2B* were generated in Fiji with adjustments of brightness and contrast.

## Microdialysis

Microdialysis was performed according to previously established methods (*Liu et al., 2018*). The probes (6 kDa MW cut-off, CMA 11, Harvard Apparatus, Catalogue# CMA8309581) were calibrated with freshly made dopamine solutions (0, 4 and 8 $\mu M$) dissolved in ACSF before each experiment. A fresh probe was used for each mouse. The microdialysis probe was continuously perfused with ACSF containing (in mM): 155 NaCl, 1.2 MgCl$_2$, 2.5 KCl, 1.2 CaCl$_2$, and 5 glucose at a speed of 1 $\mu l$/min. After probe calibration, the probe was inserted into dorsal striatum (coordinates: 1.0 mm anterior, 2.0 mm lateral of bregma, and 3.3 mm below pia) of anesthetized male and female mice using stereotaxy (73–103 days old). Striatal dialysates were collected every 15 min and the concentration of dopamine was measured using an HPLC (HTEC-510, Amuza Inc) connected to an electrochemical detector (Eicom). The data during the first 75 min were not plotted because during this time window dopamine levels stabilize after surgery. Average dopamine levels from the 76$^{th}$ - 120$^{th}$ min of Syt-1 control mice were used to normalize all dopamine values. 10 $\mu M$ TTX dissolved in ACSF was applied using reverse dialysis starting at 121 min to inhibit firing of dopamine axons as described before (*Liu et al., 2018*). For all microdialysis data acquisition and analyses, the experimenter was blind to the genotype of the mice.

## Statistical analyses

Data are expressed as mean ± SEM. All statistical analyses were performed in Graphpad Prism. Student's unpaired t-tests were used in *Figure 2B and C*, paired t-test in *Figure 1—figure supplement 1D*, Wilcoxon tests in *Figure 3B and C*, Mann Whitney tests in *Figures 1G*, *2H, I*, *3D and E*, one-way ANOVA followed by Dunnett's multiple comparisons in *Figure 1C and D* and *Figure 1—figure supplement 2C*, one-way ANOVA followed by Sidak's multiple comparisons tests in *Figure 2D and E* and *Figure 1—figure supplement 3D*, two-way ANOVA followed by Sidak's multiple comparisons

tests in *Figures 1I* and *3G*, *Figure 1—figure supplement 3E* and *Figure 1—figure supplement 4C*, and two-way ANOVA mixed-effects analysis followed by Sidak's multiple comparisons tests in *Figure 1—figure supplement 3G* and *Figure 1—figure supplement 4E*. For all genotype comparisons, the experimenter was blind to genotype during data acquisition and analyses.

## Acknowledgements

This work was supported by the National Institutes of Health (R01NS103484 to PSK), the Dean's Initiative Award for Innovation (to PSK), a Harvard-MIT Joint Research Grant (to PSK), William Randolph Hearst (to A B), Alice Joseph Brooks (to A B) and Gordon family (to CL) postdoctoral fellowships, and a Marshall Plan Foundation fellowship (to PN). We thank J Wang for technical assistance, L Kershberg for help with setting up synaptosome preparations, Drs. K Balakrishnan and W Regehr for insightful discussions, Dr. TC Südhof for providing the conditional Syt-1 knockout mice, and the Cell Biology Microscopy Facility and the Neurobiology Imaging Facility (supported by a NINDS P30 Core Center grant, NS072030) for availability of microscopes and advice.

## Additional information

### Funding

| Funder | Grant reference number | Author |
|---|---|---|
| National Institute of Neurological Disorders and Stroke | R01NS103484 | Pascal S Kaeser |
| Harvard Medical School | Dean's Initiative Award | Pascal S Kaeser |
| Harvard Medical School | Hearst Fellowship | Aditi Banerjee |
| Harvard Medical School | Brooks Fellowship | Aditi Banerjee |
| Harvard Medical School | Gordon Fellowship | Changliang Liu |
| Marshallplan-Jubiläumsstiftung | Exchange Scholarship | Paulina Nemcova |
| Harvard Medical School | Harvard-MIT Joint research grant | Pascal S Kaeser |

The funders had no role in study design, data collection and interpretation, or the decision to submit the work for publication.

### Author contributions

Aditi Banerjee, Conceptualization, Formal analysis, Investigation, Visualization, Methodology, Writing - original draft, Writing - review and editing; Jinoh Lee, Formal analysis, Investigation, Visualization, Writing - review and editing; Paulina Nemcova, Formal analysis, Investigation, Writing - review and editing; Changliang Liu, Resources, Methodology, Writing - review and editing; Pascal S Kaeser, Conceptualization, Formal analysis, Supervision, Funding acquisition, Writing - original draft, Project administration, Writing - review and editing

### Author ORCIDs

Aditi Banerjee https://orcid.org/0000-0003-2016-0717
Jinoh Lee https://orcid.org/0000-0002-2158-8507
Paulina Nemcova https://orcid.org/0000-0002-0323-8079
Pascal S Kaeser https://orcid.org/0000-0002-1558-1958

### Ethics

Animal experimentation: All animal experiments were performed according to institutional guidelines of Harvard University, and were in strict accordance with the recommendations in the Guide for the Care and Use of Laboratory Animals of the National Institutes of Health. The animals were handled according to protocols (protocol number IS00000049) approved by the institutional animal care and use committee (IACUC).

Decision letter and Author response

Decision letter https://doi.org/10.7554/eLife.58359.sa1
Author response https://doi.org/10.7554/eLife.58359.sa2

## Additional files

### Supplementary files
• Transparent reporting form

### Data availability
All data generated in the study are included in the figures, including individual data points whenever possible.

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
