## [Decision Letter]

**Acceptance summary:**

Using unequivocal mouse genetics, elegant electrophysiology, optogenetics, carbon fiber amperometry, high resolution imaging and in vivo microdialysis, your work convincingly established synaptotagmin-1 as the Ca^2+^ sensor for rapid synchronous synaptic release of dopamine in striatal neurons. It has been a pleasure to read about and handle the review process for your work.

**Decision letter after peer review:**

Thank you for submitting your article "Synaptotagmin-1 is the Ca^2+^ sensor for fast striatal dopamine release" for consideration by *eLife*. Your article has been reviewed by three peer reviewers, and the evaluation has been overseen by a Reviewing Editor and Gary Westbrook as the Senior Editor. The following individuals involved in review of your submission have agreed to reveal their identity: Zhiping Pang (Reviewer #2); Pablo Castillo (Reviewer #3).

The reviewers have discussed the reviews with one another, and the Editors drafted this decision to help you prepare a revised submission in which we ask for some minor revisions. We would like to draw your attention to changes in our revision policy that we have made in response to COVID-19 (https://elifesciences.org/articles/57162). Specifically, we ask editors to accept without delay manuscripts, like yours, that they judge can stand as *eLife* papers without additional data. Thus the revisions requested below only address clarity and presentation.

Summary:

Banerjee et al. aimed to determine the Ca^2+^ sensor for fast dopamine release in striatal neurons. Despite the obvious importance of dopamine, the molecular machinery that governs release of dopamine from nerve terminals remains enigmatic. The authors used interdisciplinary techniques including unequivocal mouse genetics, elegant electrophysiology, optogenetics, carbon fiber amperometry, high resolution imaging and in vivo microdialysis to show that synaptotagmin-1 acts as a Ca^2+^ sensor for rapid exocytosis of dopaminergic synaptic vesicles in striatal neurons. Syt-1 deletion completely abolished fast synchronous release of dopamine. These terminals were capable of Syt-1-independent asynchronous release that accounted for roughly two thirds of extracellular dopamine levels. Overall, this is a straightforward study with convincing evidence establishing Syt-1 as the main Ca^2+^ sensor for fast dopamine release in striatal neurons. All three reviewers were enthusiastic about this work and had only the following minor concerns. The editors will assess your response at the time of the revision.

1) How does Syt-1 deletion affect miniature synaptic responses in these neurons?

2) How does external Ca concentration affect asynchronous dopamine release? Recordings in the Syt-1 DAT- cKO with dopamine re-uptake blocker provides an unique opportunity to explore this a bit further.

3) The authors mention that Syt-7 levels are also high in midbrain dopaminergic neurons. Could Syt-7 be the Ca^2+^ sensor for asynchronous dopamine release?

4) Puffing of 100mM KCl in Figure 2 F-H. The authors showed that puffing high concentrations of KCl induced similar amount of dopamine release in dopamine neuronal conditional Synaptotagmin -1 knockout and in control. Can the authors elaborate on the nature of dopamine release induced by high KCl^-^mediated depolarization. See also point 9.

5) For each manipulation of extracellular Ca^2+^, was extracellular Mg^2+^ appropriately adjusted?

6) Regarding the Syt-1 cKO DA slices, the traces in Figures 1H and 3A and the normalization in 1I indicate a slow accumulation of dopamine over time as a result of optical stimulation. The Syt-1 cKO DA dopamine traces and normalization yielded from electrical stimulation (Figure 1—figure supplement 4 D,E) show a much more stable dopamine level closer to 0. What is the potential explanation for this discrepancy?

7) Figure 3C: The legend indicates that these measurements approximate total dopamine accumulated by integrating the dopamine curves between 0 and 2.9 s following the 1st stimulus. It is therefore unclear why the baseline DA level in Syt-1 cKO DA slices is exactly at 0 when the traces in Figures 1H and 3A, as well as the normalization in 1I, show a slight accumulation of dopamine over time in these slices.

8) Figure 3G: In the pre-TTX condition, the average dopamine level in Syt-1 cKO DA animals is significantly smaller than that in Syt-1 control animals at only 1 of 3 time points. Can the author provide an explanation?

9) The authors' explanation for how KCl but not ChR2 is able to cause DA release in Syt-1 cKO DA slices is not clear. Is this high KCl^-^induced release calcium-dependent?

10) The third paragraph of the main section seems to have been misplaced. The information at the start of the paragraph would help clarify rationale if presented sooner.

---

## [Author Response]

All three reviewers were enthusiastic about this work and had only the following minor concerns. The editors will assess your response at the time of the revision.1) How does Syt-1 deletion affect miniature synaptic responses in these neurons?

This is an important question in at least two ways. First, it has been shown that for conventional synaptic vesicle exocytosis, knockout of the fast Ca^2+^ sensor often increases miniature release (for key examples, see Broadie et al., 1994; Xu et al., 2009). Second, and as we discuss in the paper, there appears to be a significant amount of extracellular dopamine in the extracellular space even after removal of action-potential triggered dopamine release. This is true for knockouts for synaptotagmin, for knockouts of RIM, and for wild type mice after reverse dialysis of TTX (Figures 1 and 3 in this manuscript and Liu et al., 2018). In microdialysis experiments in the striatum, however, there is no robust increase in extracellular dopamine in Syt1 cKO^DA^ mice after TTX, suggesting that miniature release is not enhanced, or that dopamine clearance is efficient enough to counteract such an increase. Unfortunately, there is currently no reliable direct measurement for miniature dopamine release in the striatum. Notably, in the midbrain, individual spontaneous events can be detected via measuring GIRK-IPSCs (Gantz et al., 2013) and these events are unaffected after knockout of RIM (Robinson et al., 2019).

We have revised the manuscript to better explain what is known about miniature release in the dopamine system. We now state in the manuscript: “It is not known whether synaptotagmin-1 knockout affects spontaneous dopamine release, but literature from conventional synapses establishes that synaptotagmin-1 knockout strongly enhances miniature synaptic vesicle release (Broadie et al., 1994; Xu et al., 2009). Hence, it is possible that Syt-1 cKO^DA^ leads to increased spontaneous dopamine release in the striatum. But the observation that extracellular dopamine levels after TTX are not strongly increased suggests that there is no dramatic enhancement of miniature dopamine release, or that an enhancement is counteracted by dopamine clearance.”

2) How does external Ca concentration affect asynchronous dopamine release? Recordings in the Syt-1 DAT- cKO with dopamine re-uptake blocker provides an unique opportunity to explore this a bit further.

We thank the reviewers for bringing up the [Ca^2+^ ]_ex_ dependence of asynchronous dopamine release, and realize that we did not properly introduce the topic of asynchronous release in the text. While there is a large body of literature that establishes complex but positive correlations between [Ca^2+^ ]_ex_ and asynchronous release (reviewed in Kaeser and Regehr, 2014), we currently do not know what the [Ca^2+^ ]_ex_ dependence of asynchronous dopamine release is. The strong prediction from conventional synapses is that asynchronous dopamine release is positively correlated with [Ca^2+^ ]_ex_, but characterization beyond this point will require better measurements than accumulation during stimulus trains upon DAT blockade. We hope that recent developments in measurements of dopamine release will allow in the future to detect individual secretory events and their timing relative to action potential firing during trains. To better introduce asynchronous release, we added the following statement: “These additional sensors may for example mediate asynchronous release, a form of vesicular exocytosis at synapses that is triggered with a longer, variable delay in response to action potentials and Ca^2+^ entry (Kaeser and Regehr, 2014; Pang and Sudhof, 2010).”

3) The authors mention that Syt-7 levels are also high in midbrain dopaminergic neurons. Could Syt-7 be the Ca^2+^ sensor for asynchronous dopamine release?

We agree with the reviewers that, in the long-term, it will be important to know which other Ca^2+^ sensors contribute to dopamine release. Synaptotagmin-7 is indeed a leading candidate because it is expressed in these neurons. But the properties of asynchronous dopamine release are not understood, and better measurements need to be established before mechanisms of asynchronous release can be studied. Furthermore, while Synaptotagmin-7 is expressed in these neurons, it is uncertain whether Synaptotagmin-7 is present in their axonal arbor in the striatum and previous literature suggests that Synaptotagmin-7 mediates somatodendritic release (Mendez et al., 2011). We now state in the revised manuscript that synaptotagmin-7 is a candidate sensor for asynchronous release based on the literature: “While sensors for asynchronous dopamine release are not known, the presence of Synaptotagmin-7 in substantia nigra dopamine neurons and its role in asynchronous release at fast synapses (Bacaj et al., 2015; Lein et al., 2007; Mendez et al., 2011; Saunders et al., 2018; Wen et al., 2010) makes Synaptotagmin-7 a candidate sensor protein.”

4) Puffing of 100mM KCl in Figure 2 F-H. The authors showed that puffing high concentrations of KCl induced similar amount of dopamine release in dopamine neuronal conditional Synaptotagmin -1 knockout and in control. Can the authors elaborate on the nature of dopamine release induced by high KCl-mediated depolarization. See also point 9.

We thank the reviewers for pointing this out and now elaborate in the manuscript on mechanisms of KCl depolarization-induced dopamine release. The KCl puff causes massive depolarization in all neurons in the area of the puff, including dopamine axons and cholinergic interneurons (which stimulate dopamine release via nAChRs on dopamine axons). Most likely, this depolarization triggers Ca^2+^ entry into dopamine axons, which results in dopamine release. While we do not know how Ca^2+^ triggers KCl-mediated dopamine release, we hypothesize that other Ca^2+^ sensors account for it in Syt-1cKO^DA^ mice. We are confident that KCl triggers dopamine release through vesicular exocytosis at active zone-like release sites, because in mutants in which we knockout RIM from dopamine neurons, KCl-triggered dopamine release is abolished (Liu et al., 2018). We have added the following statement to the paper: “Local puffing of KCl onto brain slices causes a strong depolarization of dopamine axons and surrounding neurons, for example cholinergic interneurons, which triggers massive dopamine release that requires the active zone protein RIM in dopamine axons (Liu et al., 2018).”

Additional explanations that are related to this point are provided further below in response to point 9.

5) For each manipulation of extracellular Ca^2+^, was extracellular Mg^2+^ appropriately adjusted?

Yes, extracellular Mg^2+^ was adjusted accordingly in all experiments. The revised Materials and methods now state: “Extracellular magnesium was adjusted for recordings in variable [Ca^2+^]_ex_ (solutions contained Ca^2+^/Mg^2+^ in mM: 0.5/2.8; 1/2.3; 2/1.3 and 4/0).”

We note that we have done experiments increasing [Ca^2+^]_ex_ before without adjusting Mg^2+^ (Liu et al., 2018), and that the increase in dopamine release was similar to what we report here.

6) Regarding the Syt-1 cKO DA slices, the traces in Figures 1H and 3A and the normalization in 1I indicate a slow accumulation of dopamine over time as a result of optical stimulation. The Syt-1 cKO DA dopamine traces and normalization yielded from electrical stimulation (Figure 1—figure supplement 4 D,E) show a much more stable dopamine level closer to 0. What is the potential explanation for this discrepancy?

The key difference between the two experiments is that in the main figures (Figures 1H and 3A), optogenetic stimulation was used while in the figure supplement (Figure 1—figure supplement 4D and 4E) electrical stimulation was applied. Optogenetic stimulation only recruits dopamine axons and leads to depression. In contrast, electrical stimulation recruits dopamine axons and cholinergic interneurons, and stimulating cholinergic interneurons strongly enhances depression (Liu et al., 2018; Threlfell et al., 2012; Zhou et al., 2001). Hence, the discrepancy is explained by recruitment of the cholinergic mechanism and is consistent with previous papers published on this topic. We now state in the figure legend of Figure 1—figure supplement 4: “Electrical stimulation activates dopamine fibers and cholinergic interneurons, and cholinergic innervation of dopamine axons accounts for as much as ~90% of the extracellular dopamine detected upon electrical stimulation and leads to enhanced depression of dopamine release during stimulus trains (Liu et al., 2018; Threlfell et al., 2012; Zhou et al., 2001).”

7) Figure 3C: The legend indicates that these measurements approximate total dopamine accumulated by integrating the dopamine curves between 0 and 2.9 s following the 1st stimulus. It is therefore unclear why the baseline DA level in Syt-1 cKO DA slices is exactly at 0 when the traces in Figures 1H and 3A, as well as the normalization in 1I, show a slight accumulation of dopamine over time in these slices.

We thank the reviewers for bringing this up and would like to note that the baseline DA level in Syt1 cKO^DA^ was not exactly zero, but the scale of the y-axis was such that it was difficult to see this. We have now split up the graph in Figure 3C such that Syt1 control and Syt1 cKO^DA^ values are shown on separate scales and the low amount of accumulation at baseline can be visualized. We thank the reviewers for pointing this out.

8) Figure 3G: In the pre-TTX condition, the average dopamine level in Syt-1 cKO DA animals is significantly smaller than that in Syt-1 control animals at only 1 of 3 time points. Can the author provide an explanation?

In our view, the pre-TTX effect of Syt1 cKO^DA^ in Figure 3G is surprisingly mild given the strong reduction in synchronous release. Because the pre-TTX effect of Syt1 cKO^DA^ is mild and there is some variability in microdialysis measurements, it is difficult to detect. This is particularly striking because in mutants that lack RIM in dopamine neurons in which action potential triggered release is also absent, pre-TTX dopamine levels in microdialysis are very strongly reduced (Liu et al., 2018).

We describe the effect of synaptotagmin-1 ablation on pre-TTX dopamine in vivo as follows : “Remarkably, extracellular dopamine levels before TTX reverse dialysis were only mildly reduced in Syt-1 cKO^DA^…”.

While we don’t have a definitive answer as to why this is the case, we think it is most likely that in vivo, asynchronous release contributes robustly to extracellular dopamine because dopamine levels in microdialysis drop significantly after action potential blockade with TTX in Syt-1 cKO^DA^ mice, and changes in the clearance of dopamine could also contribute. We also note that there is a good match across different mutants between dopamine in the microdialysate and KCl depolarization induced release: in Syt1 cKO^DA^ animals, amperometric currents in response to action potentials are strongly reduced, but KCl depolarization and in vivo microdialysis reports significant extracellular dopamine. This is different from dopamine neuron RIM knockouts (RIM cKO^DA^), in which KCl triggered release is abolished and pre-TTX extracellular dopamine in microdialysis is strongly reduced (Liu et al., 2018). To summarize these effects, we now state the following: “These data suggest that three modes of dopamine release exist: synchronous and asynchronous release in response to action potentials, and action potential independent release that is likely mediated by spontaneous exocytotic events. Remarkably, each component appears to contribute significantly to the extracellular dopamine measured by microdialysis in anesthetized mice, suggesting that asynchronous and action potential-independent release may be prominent.”

We hope that these statements, together with the explanations above, address this concern.

9) The authors' explanation for how KCl but not ChR2 is able to cause DA release in Syt-1 cKO DA slices is not clear. Is this high KCl^-^induced release calcium-dependent?

We thank the reviewers to bring up this important point, and it relates to point 4 as well. This point has two aspects: (1) how channelrhodopsin triggers release, and (2) how KCl triggers release.

1) While it is possible that Ca^2+^ enters through channelrhodopsin directly under some circumstances, this is not the case here in a way that Ca^2+^ is sufficient to trigger dopamine release. The experiments in Syt1 cKO^DA^ animals are performed with oChiEF, not ChR2, and we have directly tested whether sodium channels are necessary for this form of release. We found that TTX entirely blocks oChiEF induced release (Figures 6A-6C in Liu et al., 2018). Hence, this method triggers release by evoking action potentials in dopamine axons. We now state: “In these mice, we expressed oChiEF-citrine, a fast channelrhodopsin, selectively in dopamine neurons using AAVs (Figure 1A) to optogenetically evoke dopamine release through triggering of axonal action potentials (Liu et al., 2018).”

2) As outlined above, KCl triggers release most likely through at least two mechanisms, massive depolarization of (i) dopamine axons and (ii) of cholinergic interneurons, which in turn release acetylcholine and then trigger dopamine release through an unknown mechanism after activation of dopamine axonal nAChRs. Notably, this form of dopamine release requires the presence of vesicular release machinery, for example RIM, and hence we concluded in a previously published study that it is vesicular (Liu et al., 2018). Because release triggered through nAChRs activation (Figure 1—figure supplement 4) is abolished in Syt1 cKO^DA^ mice, we conclude that KCl triggers vesicular exocytosis of dopamine through strong depolarization of dopamine axons.

We now specifically describe the mechanisms of KCl mediated release: “Local puffing of KCl onto brain slices causes a strong depolarization of dopamine axons and surrounding neurons, for example cholinergic interneurons, which triggers massive dopamine release that requires the active zone protein RIM in dopamine axons (Liu et al., 2018).”

The best experiment to directly test the requirement for extracellular Ca^2+^ for this form of release would be to remove extracellular Ca^2+^ entirely. For optogenetic experiments, we have tried to fully remove extracellular Ca^2+^ from striatal slices, but have found that this is challenging and we do not have good experiments to evaluate the requirement of extracellular Ca^2+^ for KCl^-^triggered dopamine release. Hence, although there is a long-term body of literature to suggest that KCl triggers release through depolarization-dependent opening of voltage gated Ca^2+^ channels followed by flooding of the nerve terminal with Ca^2+^, we do not have the capability to prove this at this point in striatal brain slices. To account for this in the manuscript, we now state: “While the exact mechanism of KCl depolarization induced dopamine release is not known, the data further suggest that depolarization-induced, likely massive Ca^2+^ entry may trigger vesicular dopamine release via one or multiple alternative Ca^2+^ sensors in the absence of Synaptotagmin-1.”

10) The third paragraph of the main section seems to have been misplaced. The information at the start of the paragraph would help clarify rationale if presented sooner.

We thank the reviewers for this stylistic suggestion and have switched the paragraphs as suggested to present some of the rationale sooner.

References

Gantz, S.C., Bunzow, J.R., and Williams, J.T. (2013). Spontaneous Inhibitory Synaptic Currents Mediated by a G Protein-Coupled Receptor. Neuron *78*, 807–812.

Robinson, B.G., Cai, X., Wang, J., Bunzow, J.R., Williams, J.T., and Kaeser, P.S. (2019). RIM is essential for stimulated but not spontaneous somatodendritic dopamine release in the midbrain. e*Life8*.